# Dissecting the DNA binding landscape and gene regulatory network of p63 and p53

Konstantin Riege[1], Helene Kretzmer[2], Arne Sahm[1], Simon S McDade[3], Steve Hoffmann[1], Martin Fischer[1]*

[1]Computational Biology Group, Leibniz Institute on Aging – Fritz Lipmann Institute (FLI), Jena, Germany; [2]Department of Genome Regulation, Max Planck Institute for Molecular Genetics, Berlin, Germany; [3]Patrick G Johnston Centre for Cancer Research, Queen's University Belfast, Belfast, United Kingdom

**Abstract** The transcription factor p53 is the best-known tumor suppressor, but its sibling p63 is a master regulator of epidermis development and a key oncogenic driver in squamous cell carcinomas (SCC). Despite multiple gene expression studies becoming available, the limited overlap of reported p63-dependent genes has made it difficult to decipher the p63 gene regulatory network. Particularly, analyses of p63 response elements differed substantially among the studies. To address this intricate data situation, we provide an integrated resource that enables assessing the p63-dependent regulation of any human gene of interest. We use a novel iterative de novo motif search approach in conjunction with extensive ChIP-seq data to achieve a precise global distinction between p53-and p63-binding sites, recognition motifs, and potential co-factors. We integrate these data with enhancer:gene associations to predict p63 target genes and identify those that are commonly de-regulated in SCC representing candidates for prognosis and therapeutic interventions.

*For correspondence:
Martin.Fischer@leibniz-fli.de

**Competing interests:** The authors declare that no competing interests exist.

## Introduction

In contrast to the tumor suppressor p53 with its extensive set of target genes controlling the cell cycle and apoptosis (*Fischer, 2017*; *Sammons et al., 2020*), its phylogenetically ancient sibling p63 (ΔNp63) governs epidermis development (*Mills et al., 1999*; *Yang et al., 1999*) and is an oncogenic driver of squamous cell carcinoma (SCC) (*Cancer Genome Atlas Research Network et al., 2018*; *Gatti et al., 2019*) that is overexpressed or amplified in SCCs, which depend on its expression (*Ramsey et al., 2013*). Together with p73, p63, and p53 form the p53 transcription factor (TF) family that shares a highly conserved DNA binding domain (DBD) through which they bind to very similar DNA recognition motifs. The mechanisms that enable these sibling TFs to shape their unique gene regulatory network (GRN) leading to the different phenotypic control, however, remain poorly understood.

The *TP53* and *TP63* genes encode for two major isoform groups that are controlled by distinct promoters leading to transcripts differing in their N-terminus (*Murray-Zmijewski et al., 2006*). In the case of *TP53*, the longest isoform, p53α, is ubiquitously expressed while the alternative intronic promoter has little activity across virtually all tissues. Conversely, the usage of the two *TP63* promoters is highly cell type-dependent. For instance, the long isoform TAp63 is predominantly expressed in germ cells, while the smaller transcript, ΔNp63, is most copious in stratifying epithelia (*Sethi et al., 2015*). Similar to p53, alternative splicing leads to α, β, and γ protein isoforms that differ in their C-terminus (*Murray-Zmijewski et al., 2006*). While both TAp63 and ΔNp63 may bind to DNA through a specific binding domain, ΔNp63 lacks the canonical N-terminal transactivation

domain (TAD) (*Yang et al., 1998*) and has long been thought to be a dominant-negative regulator of other p53 family members or its own isoforms (*Gebel et al., 2016*; *Yang et al., 1998*). However, ΔNp63 has also been shown to harbor alternative TADs, that endow transactivation activity (*Helton et al., 2006*; *King et al., 2003*; *Yang et al., 2006*). Notably, many ΔNp63-binding sites are associated with enhancer regions, where ΔNp63 has been proposed to 'bookmark' genes that are expressed in stratifying epithelia (*Karsli Uzunbas et al., 2019*; *Kouwenhoven et al., 2015a*; *Lin-Shiao et al., 2019*; *Qu et al., 2018*; *Somerville et al., 2018*). Here, we focus on the most widely expressed isoforms p53α (hereafter p53) and ΔNp63 (hereafter p63).

The p53 TF family shares many binding sites, but all three family members have been shown to bind to substantial sets of unique target genes (*Lin et al., 2009*; *McDade et al., 2014*). Indeed, there are differences in the DBDs, for example regarding thermostability, hydrophobic potentials (*Enthart et al., 2016*), zinc-coordination (*Lokshin et al., 2007*), and redox sensitivity (*Tichý et al., 2013*). In addition, the different C-terminal domains (CTD) of p53 family members may also affect their DNA binding specificity (*Sauer et al., 2008*). p53 binds to a canonical 20 bp response element (RE) made of two decameric half-sites that both contain the sequence RRRCWWGYYY (R = A/G; W = A/T; Y = C/T). p53 has also been shown to bind to decameric half-sites separated by spacers or to single half-sites (*Kitayner et al., 2010*; *Menendez et al., 2013*; *Vyas et al., 2017*). Results from systematic evolution of ligands by exponential enrichment (SELEX) (*Ortt and Sinha, 2006*; *Perez et al., 2007*) and high-throughput analyses of chromatin immunoprecipitation (ChIP) (*Kouwenhoven et al., 2010*; *McDade et al., 2012*; *Yang et al., 2006*) yielded p63 binding motifs with high similarity to the p53RE but still showed some unique characteristics. These unique characteristics identified for p63REs, however, differed substantially between the studies.

While multiple genome-wide p63 gene expression datasets became available in recent years, our understanding of the p63 GRN remains incomplete. This is in part due to the limited overlap of the p63-dependent genes identified in individual studies (*Kouwenhoven et al., 2015b*). Also, the frequent binding of p63 to enhancers (*Kouwenhoven et al., 2015a*; *Lin-Shiao et al., 2019*; *Lin-Shiao et al., 2018*; *Qu et al., 2018*; *Somerville et al., 2018*) and the difficulty to associate such enhancers with target gene regulation adds another level of complexity to the quest of describing the GRN. To overcome these limitations, we utilize a recently developed meta-analysis approach (*Fischer et al., 2016a*), which helped us to dissect the GRNs of the mouse and human orthologue of p53 (*Fischer, 2020*; *Fischer, 2019*). The analysis rests upon a ranking of potential p63 target genes based on the number of datasets supporting a p63-dependent regulation. In addition, we utilize the wealth of recent p63 and p53 ChIP-seq studies to establish a more precise global distinction between p53- and p63-binding sites and their underlying REs. This approach could serve as a blueprint to distinguish binding site specificities of TF siblings. Further integration of gene expression studies with the binding data and enhancer:gene associations enables us to predict high-probability direct p63 target genes.

## Results

### The p63 gene regulatory network

To identify genes commonly regulated by p63 across cell types and tissues, we employed a previously established meta-analysis approach, that has been helpful to infer core GRNs for human and mouse p53, the viral oncoprotein E7 and the cell cycle GRN (*Fischer, 2019*; *Fischer et al., 2017*; *Fischer et al., 2016a*; *Fischer et al., 2014*). From 11 genome-wide studies (*Abraham et al., 2018*; *Bao et al., 2015*; *Carroll et al., 2006*; *Gallant-Behm et al., 2012*; *Karsli Uzunbas et al., 2019*; *Lin-Shiao et al., 2019*; *Saladi et al., 2017*; *Somerville et al., 2018*; *Watanabe et al., 2014*; *Wu et al., 2012*; *Zarnegar et al., 2012*; *Supplementary file 1*), 16 publically available gene expression datasets were integrated to generate a specific *p63 Expression Score* (*Supplementary file 2*). The datasets have been obtained from knockdown (n = 12) or overexpression experiments (n = 4) of p63 in primary keratinocytes (n = 3), the keratinocyte cell line HaCaT (n = 2), the foreskin fibroblast cell line BJ (n = 1), the breast epithelial cell line MCF10A (n = 4), the squamous carcinoma cell lines H226 (n = 2), KYSE70 (n = 1), and FaDu (n = 1), as well as the pancreatic ductal adenocarcinoma cell lines BxPC3 (n = 1) and SUIT2 (n = 1) (*Figure 1A and B* and *Supplementary file 1*).

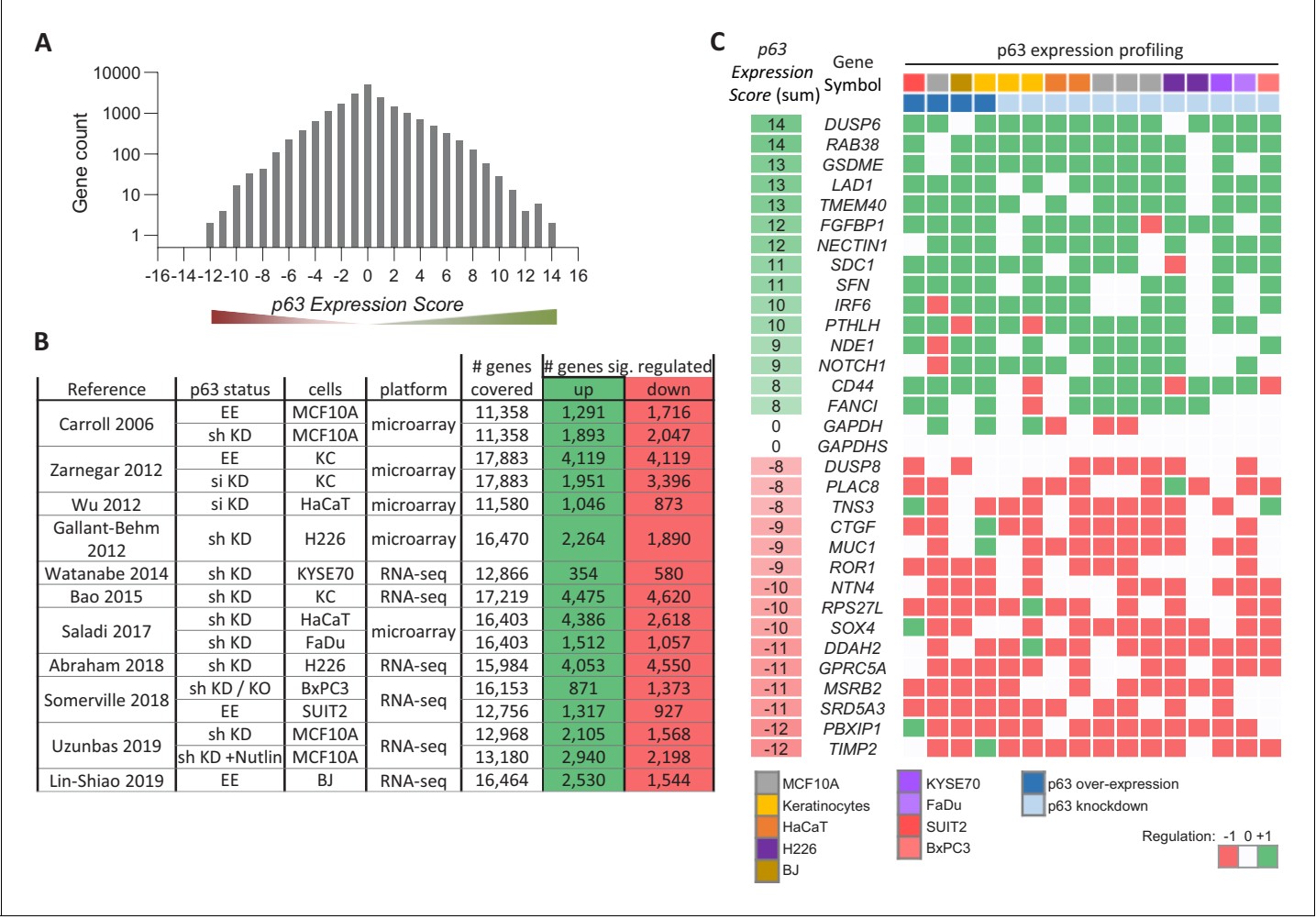

**Figure 1.** Meta-analysis of p63-dependent gene regulation. (A) Distribution of the number of genes found in each of the *p63 Expression Score* groups. Because *p63 Expression Score* group '14' and '−12' contained only two genes they were included in group '13' and '−11', respectively, for further analyses. (B) 16 datasets on p63-dependent gene expression from 11 studies. EE – exogenous p63 expression; sh KD – shRNA-mediated knockdown; si KD – siRNA-mediated knockdown; KO - sgRNA-mediated knockout (C) A heatmap displaying the regulation of 15 genes with positive and 15 genes with negative *p63 Expression Scores*. GAPDH and GAPDHS represent negative controls.

To illustrate the utility of our approach, we selected 30 genes from various *p63 Expression Score* groups reflecting commonly up- and down-regulated ones (*Figure 1C*). We noted lower consistency across the data on p63-dependent gene regulation as compared to previous meta-analyses on human and mouse p53 (*Fischer, 2019*; *Fischer et al., 2016a*). In contrast to the recent investigations, data integrated here are based on a higher number of experiments in primary cells and a comparably lower number of replicates. Thus, the reduced consistency may also reflect the higher variance as opposed to data from more homogenous cell lines. Furthermore, p63-depleted cells are less viable, and the global decrease in mRNA levels may confound effects. Despite this, our approach identified genes that are commonly altered by p63.

We next performed gene set enrichment analysis (GSEA) for p63-dependently regulated genes using MSigDB gene sets (*Subramanian et al., 2005*). In agreement with the function of p63 as an essential proliferation factor (*McDade et al., 2011*; *Senoo et al., 2007*; *Truong et al., 2006*), epidermal development regulator (*Mills et al., 1999*; *Yang et al., 1999*), and MYC network activator (*Wu et al., 2012*), we find that genes commonly up-regulated by p63 significantly enrich gene sets associated with cell cycle, epidermis development, and MYC targets (*Figure 2A*). In line with previous reports (*Mehta et al., 2018*), genes down-regulated by p63 enrich gene sets connected with interferon response (*Figure 2B*). Corroborating the role of p63 in mammary stem cell activity

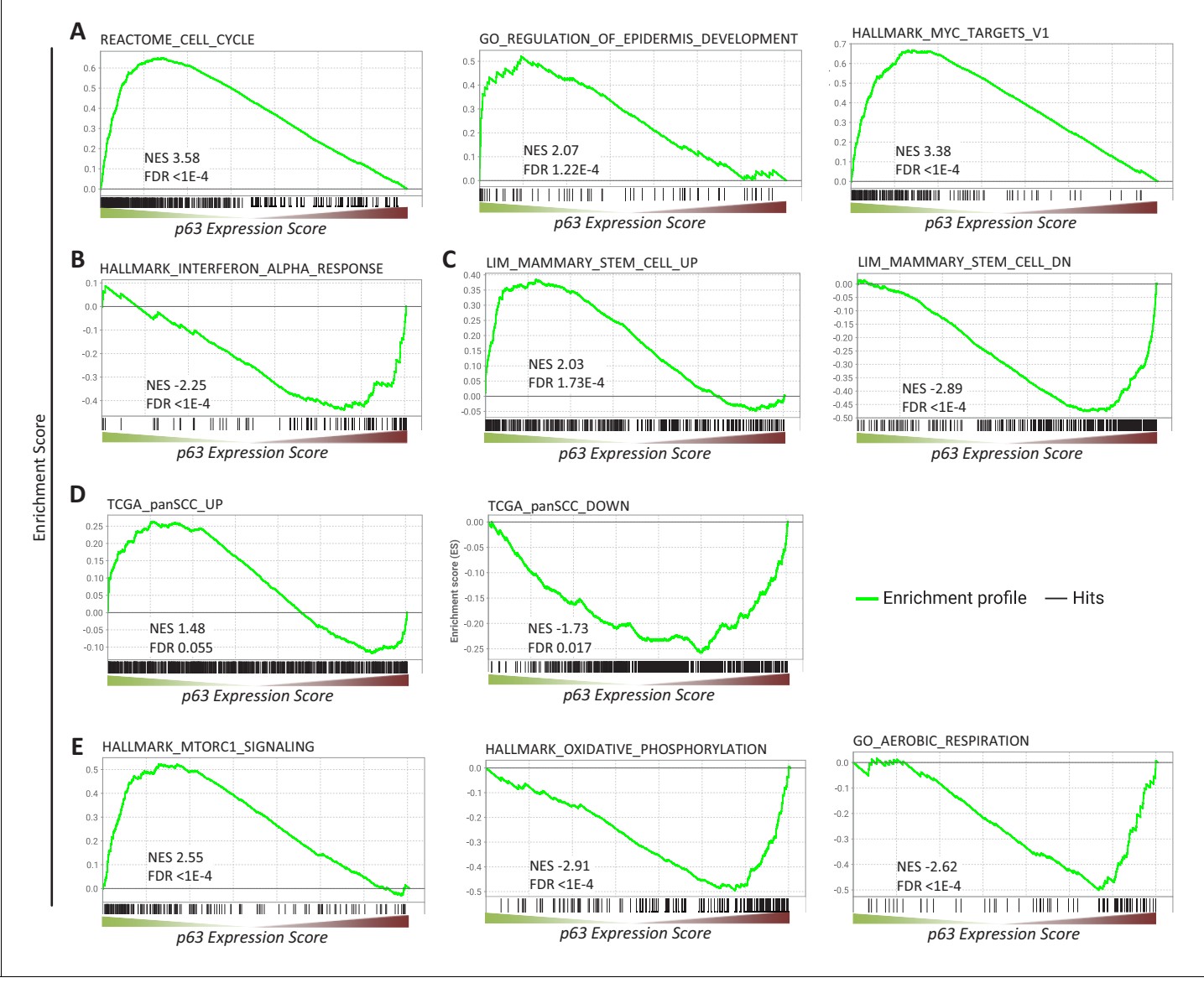

**Figure 2.** Gene sets enriched among genes commonly regulated by p63. Enrichment of (**A, B, C, E**) MSigDB gene sets or (**D**) genes up- and down-regulated across squamous cell cancers (SCC) (*Cancer Genome Atlas Research Network et al., 2018*) among genes ranked by the *p63 Expression Score*.

(*Chakrabarti et al., 2014*) and SCC growth (*Ramsey et al., 2013*), we find that p63 up- and down-regulated genes enrich respective gene sets up- and down-regulated in mammary stem cells (*Figure 2C*) and across SCCs (*Figure 2D*). In addition to pathways that have been linked to p63 earlier, we find that p63 up-regulated genes enrich for mTORC1 signaling genes and p63 down-regulated genes enrich for gene sets associated with oxidative phosphorylation and aerobic respiration (*Figure 2E*).

Further, we performed TF binding enrichment analysis for p63-dependently regulated genes using Enrichr (*Kuleshov et al., 2016*). In agreement with its established roles, we identify cell cycle gene regulators (E2F4, E2F6, SIN3A, E2F1, FOXM1, NFYA, and NFYB *Fischer and Müller, 2017*) and the MYC/MAX TFs as being enriched among p63-upregulated genes. Consistent with previous reports, our analysis also identifies KLF4 (*Sen et al., 2012*) and SMAD4 (*Rodriguez Calleja et al., 2016*) as potential mediators of p63-dependent gene regulation. In addition, our analysis reveals that androgen receptor (AR), its co-factor ZMIZ1, as well as SP1, FLI1, and NANOG are novel

candidates for mediating the p63-dependent up-regulation of multiple genes. Surprisingly, our analysis identified only SOX2 as a frequent binder of genes down-regulated by p63 (*Figure 3A*). Consistent with the strong association of p63 up-regulated genes with the cell cycle (*Figure 2A*) and with cell cycle regulators (*Figure 3A*), we find that p63 up-regulated genes enrich DREAM (dimerization partner, RB-like, E2F, and multi-vulval class B) and E2F target genes (*Figure 3B*), and DREAM target genes appear to be modestly but consistently down-regulated when p63 is lost (*Figure 3C*). Notably, most datasets on p63-dependent gene expression were derived from cells in which p63 was overexpressed or depleted, without additional treatments. However, one dataset was derived from Nutlin-treated MCF10A cells (*Karsli Uzunbas et al., 2019*). MCF10A cells harbor wild-type p53 and DREAM targets are down-regulated in response to Nutlin treatment. Strikingly, depletion of p63 decreased their expression even further without affecting *CDKN1A* (p21) levels (*Figure 3D*), which indicates a possible cumulative effect that is independent of p53 regulatory functions.

Together, the meta-analysis approach overcomes the limitations of individual studies and identifies target genes supported by multiple datasets. The extensive and integrated resource on p63-regulated genes enables researchers to compare their results quickly and to identify the most promising targets.

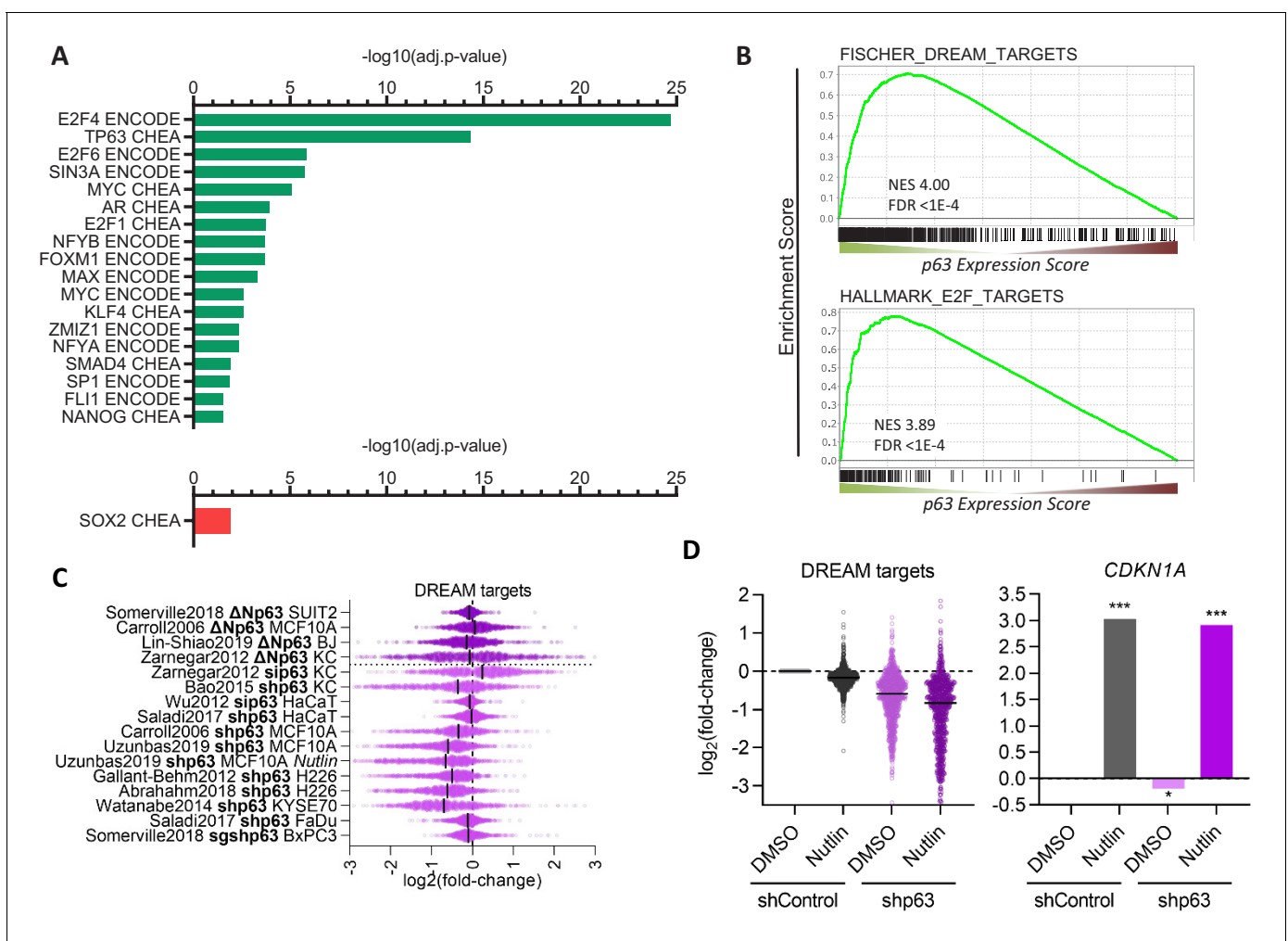

**Figure 3.** Transcription factors in the p63 GRN. (A) Significant (adj.p-value≤0.05) enrichment of TF binding at genes with a *p63 Expression Score* ≥8 (green) or ≤ −8 (red) as identified by Enrichr (*Kuleshov et al., 2016*). (B) Enrichment of MSigDB gene sets among genes ranked by the *p63 Expression Score*. Scatter plots displays the log2(fold-change) of previously collected high confidence DREAM target genes (*Fischer et al., 2016a*) (C) across the 16 p63-dependent gene expression profiling datasets and (D) MCF10A cells treated with DMSO or Nutlin in addition to shControl and shp63 (*Karsli Uzunbas et al., 2019*). *CDKN1A* levels serve as control. The black line indicates the median.

## p63 and p53 regulate largely distinct gene sets

Given that p63 and p53 share a significant number of binding sites and thus potential target genes, we next compared the *p63 Expression Score* to the previously established *p53 Expression Score* (*Fischer et al., 2016a*). In agreement with the up-regulation of cell cycle genes and DREAM targets through p63 (*Figures 2A* and *3*) and their down-regulation through p53 (*Fischer et al., 2016a*; *Fischer et al., 2016b*; *Schade et al., 2019*; *Uxa et al., 2019*), we noted that negative *p53 Expression Scores* tend to correlate with positive *p63 Expression Scores* (*Figure 4A*). Furthermore, the results indicate that p53-induced genes (positive *p53 Expression Scores*) appear to be largely unaffected by p63. Consistently, expression data for 343 target genes with strong evidence for direct up-regulation by p53 (*Fischer, 2017*), do not show consistent expression changes upon knockdown or induction of p63 (*Figure 4B*). Together, these results indicate that basal expression of the majority of p53 target genes is not affected by p63.

## Common and distinct properties of p63 and p53 DNA binding

To identify shared p63- and p53-bound sites, we compared the 20 p63 ChIP-seq datasets (*Supplementary file 1*) to 28 p53 ChIP-seq datasets we compiled recently (*Fischer, 2019*). Notably, p63 and p53 data was collected from cells with strong basal p63 expression and stimulated p53 expression, respectively. While the majority of all p53 ChIP-seq peaks occurs in only one of the experiments, more than half of the p63 peaks are present in two or more datasets (*Figure 5A and B*). Even though we were able to integrate substantially more p53 datasets, the number of identified p63 binding sites was still higher (*Figure 5C*). This indicates that p63 occupies many more binding sites as compared to p53. Importantly, when more datasets agree on p53- and p63-binding sites, these sequences are more likely to harbor a canonical p53 and p63RE, facilitating the motif discovery by tools such as HOMER (*Heinz et al., 2010*) and enriching bona fide binding sites (*Figure 5D*). Earlier meta-analyses employed a similar strategy (*Fischer et al., 2016a*; *Nguyen et al., 2018*; *Verfaillie et al., 2016*). To dissect the binding preferences of p63 and p53, we generated three distinct peak sets (*Figure 5E*). The 'p53+p63' set contained all binding sites with evidence in at least five p63 and five p53 ChIP-seq datasets. The 'p53 unique' (hereafter 'p53') set contained all binding sites that were supported by at least five p53 ChIP-seq datasets but not a single p63 dataset. We also generated a 'p63 unique' (hereafter 'p63') set vice versa.

We employed an iterative de novo motif search using HOMER to identify frequent binding site motifs. After each round, we removed all peaks harboring the best motif and repeated the search.

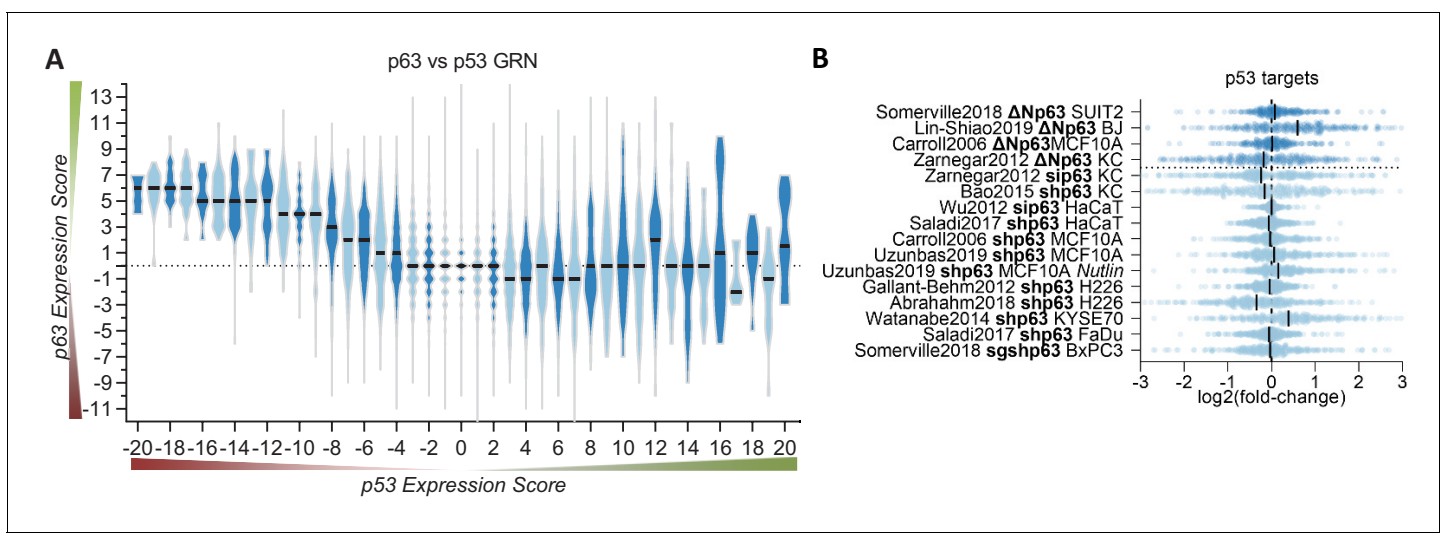

**Figure 4.** p63 and p53 regulate largely distinct target gene sets. (**A**) The *p63 Expression Score* compared to the previously published *p53 Expression Score* that was generated using the same meta-analysis approach (*Fischer et al., 2016a*) for all 16,198 genes for which both scores were available. (**B**) The scatter plot displays the log$_2$(fold-change) of previously collected high confidence direct p53 target genes (*Fischer, 2017*) across the 16 p63-dependent gene expression profiling datasets. The black line indicates the median. The data indicates a large degree of independence of p53 targets from p63-dependent expression.

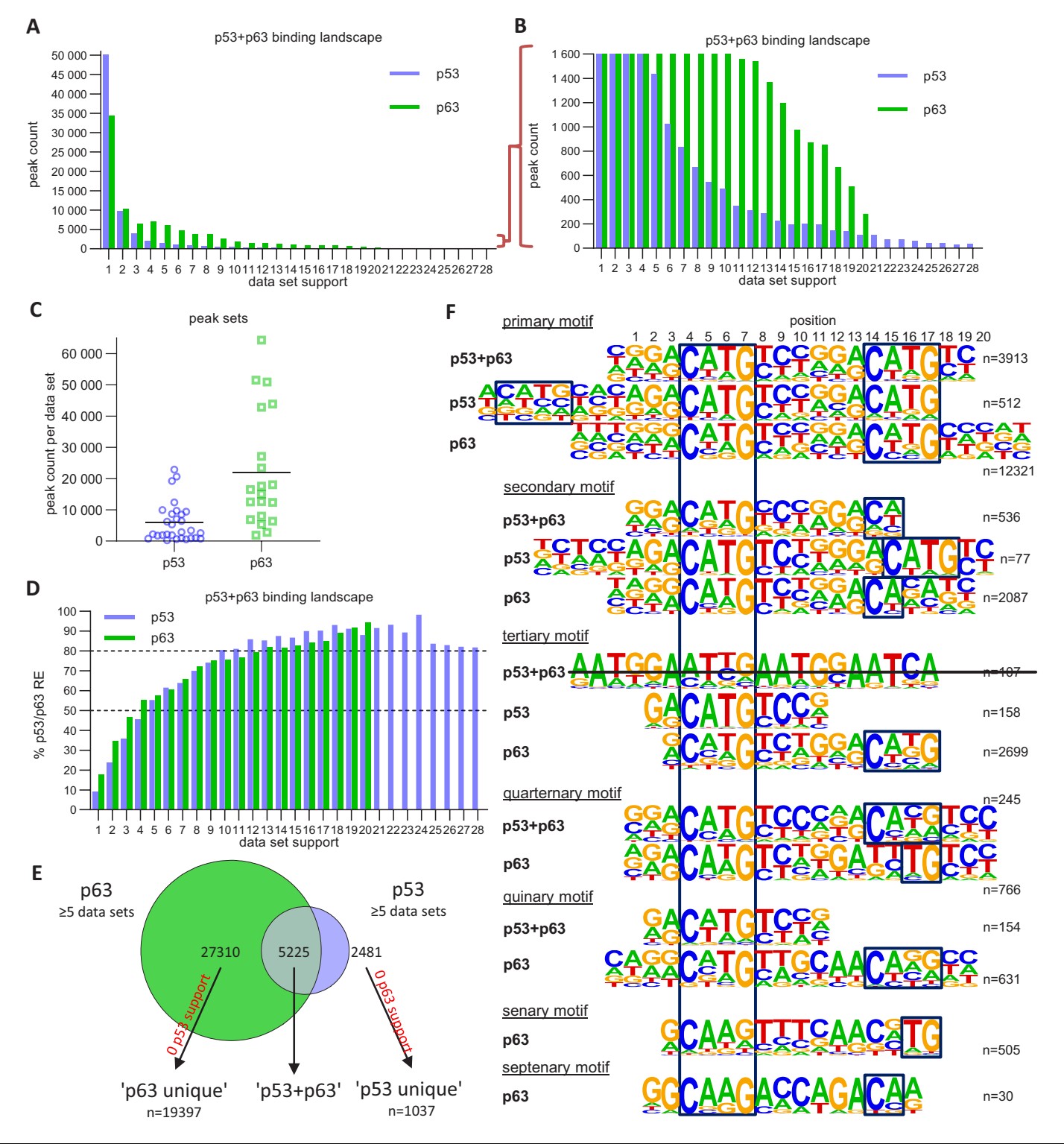

**Figure 5.** The p63 and p53 DNA-binding landscape. (**A and B**) The number of p63 and p53 binding peaks sorted by the number of datasets that commonly identified/support the peak. (**C**) The number of p53 and p63 peaks identified in the 28 p53 and 20 p63 ChIP-seq datasets, respectively. (**D**) The relative number of 'known' p53 and p63 motifs found by HOMER v4.10 (*Heinz et al., 2010*) under p53 and p63 peaks, respectively, with increasing dataset support. (**E**) Schematic of 'p53', 'p63' and 'p53+p63' peak selection for further analyses. (**F**) De novo motif search results from HOMER v4.10 for the 'p53+p63', 'p53', and 'p63' peak sets. The first round of motif search identified the 'primary' motif in each peak set. Using an iterative approach, all

*Figure 5 continued on next page*

*Figure 5 continued*

peaks that contained the 'primary' motif were removed and the de novo motif search was repeated. This iterative approach was followed until no more p53/p63-like motif was identified.

The online version of this article includes the following figure supplement(s) for figure 5:

**Figure supplement 1.** Correlation between p53 and p63 binding frequency and motif consensus.
**Figure supplement 2.** Correlation between p53 and p63 binding frequency and motif consensus.
**Figure supplement 3.** Top motifs co-enriched with primary 'p53+p63', 'p53', and 'p63' motifs at the respective DNA sites.
**Figure supplement 4.** TFs with significantly similar binding repertoirs.

We identified similar yet distinct binding motifs for the three groups (*Figure 5F*). Comparison of the primary 'p53+p63', 'p53', and 'p63' motifs suggests that p63-binding sites display a highly conserved C, G, C, and G at positions 4, 7, 14, and 17, respectively. The second round revealed a p53RE containing a 1 bp spacer (p53 secondary motif), supporting the model that p53 can bind to spacer-containing p53REs (*Vyas et al., 2017*). The results further indicate that p53 can bind to a single half-site (p53 tertiary motif) and that this single half-site is more constrained at positions 5 and 6 as well as the flanking regions than half-sites in the canonical p53RE (e.g. primary p53+p63 and p53 motifs). Of note, these single half-sites may also include p53REs with spacers longer than 1 bp that are not detected separately because of their very low abundance. Sole half-sites together with spacer-containing p53REs underlie only ~5% of p53-bound sites (*Figure 6*). Furthermore, p53 and p63 appear to be able to bind to three-quarter sites (secondary and quaternary p53+p63 and p63 motifs), while p63 can generally bind to a broader spectrum of sequences as compared to p53

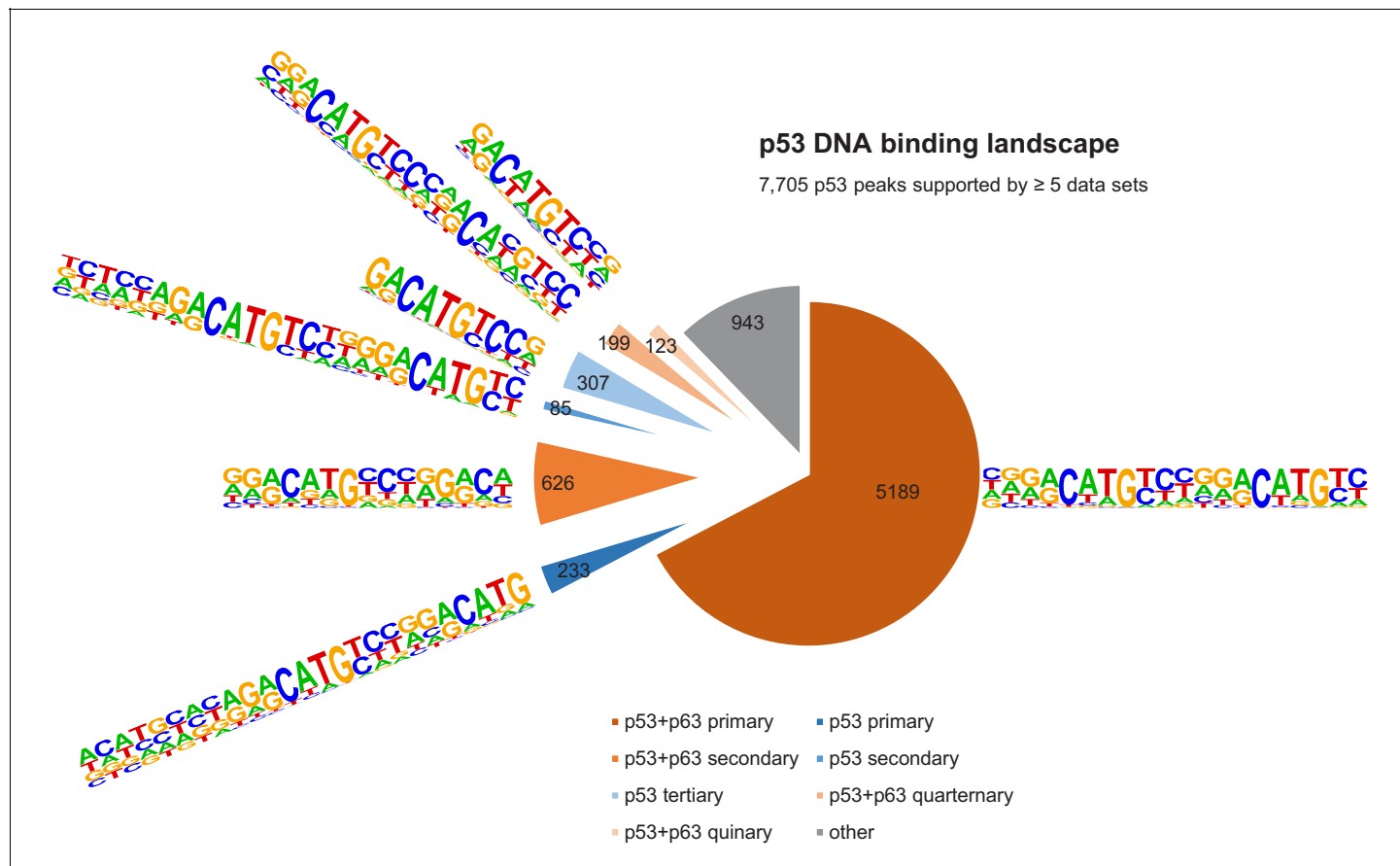

**Figure 6.** The DNA-binding landscape of p53. DNA sites occupied by p53 in at least five datasets were searched iterative with the motifs identified by our iterative de novo search (*Figure 5F*). We searched first for the primary 'p53+p63' motif and among all remaining sites for the primary 'p53' motif. All other 'p53+p63' and 'p53' motifs were searched subsequently.

(*Figure 5F*). This broader binding repertoire likely underlies p63's capacity to engage with substantially more binding sites than does p53.

It is important to note that the vast majority (~70%) of p53- and p63-binding sites harbor full-length p53 and p63REs (*Figures 6* and *7*, *Supplementary file 3*). There is a good correlation between p53- and p63-binding site occupation, and most sites commonly bound by p53 are also frequently bound by p63 (*Figure 5—figure supplement 1*). However, p63 binds many sites that are not bound by p53 (*Figure 5E* and *Figure 5—figure supplement 2*). More importantly, p53 binding is strongly constrained to canonical p53RE (*Figure 5—figure supplements 1C–D* and *2A–C*). In contrast, p63 binding appears not to benefit from a more canonical p63RE (*Figure 5—figure supplements 1E–F* and *2D–F*). These data suggest that sequence-specific binding is particularly important to recruit p53, while p63 only requires minimal sequence identity and could require additional cofactors to bind and ultimately regulate its target genes.

Therefore, we also searched for potential cooperating TFs that may be co-enriched at p53- and p63-binding sites. Consistent with earlier analyses (*Verfaillie et al., 2016*), no additional motif was substantially enriched in the vicinity of 'p53' or 'p53+p63' binding sites. Consistent with the co-enrichment of AP-1 and p63 at enhancers (*Lin-Shiao et al., 2018*), we found that unique p63-binding sites were consistently enriched for AP-1 (bZIP) in addition to bHLH motifs (*Figure 5—figure supplement 3*). Using the CistromeDB toolkit (*Zheng et al., 2019b*), we identified TFs that significantly

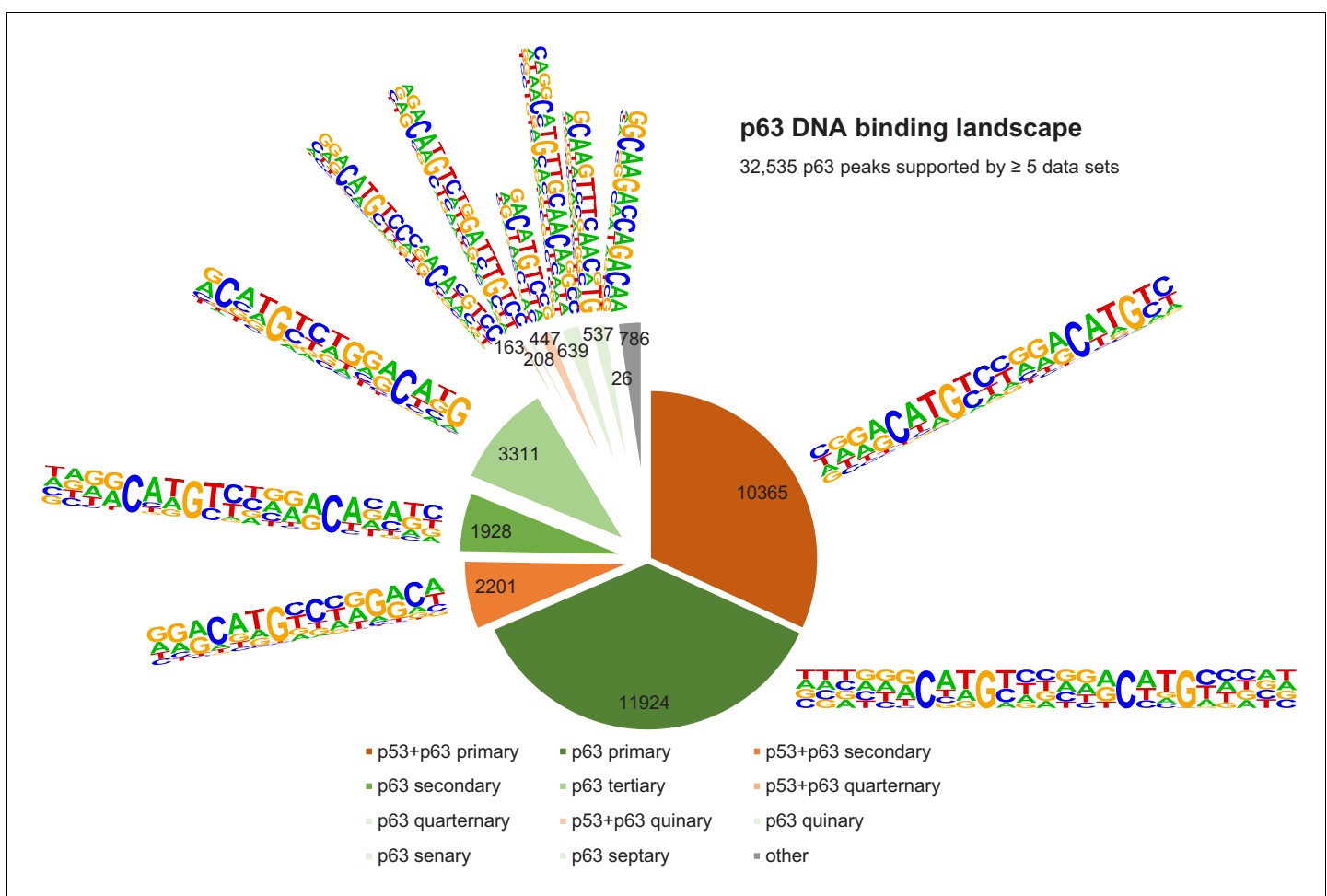

**Figure 7.** The DNA-binding landscape of p63. DNA sites occupied by p63 in at least five datasets were searched iterative with the motifs identified by our iterative de novo search (*Figure 5F*). We searched first for the primary 'p53+p63' motif and among all remaining sites for the primary 'p63' motif. All other 'p53+p63' and 'p63' motifs were searched subsequently (*Supplementary file 3*).

The online version of this article includes the following figure supplement(s) for figure 7:

**Figure supplement 1.** Complement to *Table 1*.

enrich for binding to the 'p53+p63', 'p53', and 'p63' sites. As expected, the analysis identified the p53 family members p53, p63, and p73 as best hits for the common sites, but only p53 and p73 for the unique p53 and only p63 and p73 for the unique p63 peak sets (*Figure 5—figure supplement 4*). In agreement with earlier studies, the analysis identified p300 (*Katoh et al., 2019*), MAF (*Lopez-Pajares et al., 2015*), SOX2 (*Watanabe et al., 2014*), BANF1 (also known as BAF) (*Bao et al., 2015*), and KMT2D (*Lin-Shiao et al., 2018*) as potential co-binders of p63; as well as TRIM28 (*Doyle et al., 2010*), BRD4 (*Stewart et al., 2013*), p300 (*Lill et al., 1997*), ZBTB33 (KAISO) (*Koh et al., 2014*), CDK9 (*Claudio et al., 2006*), and HEXIM1 (*Lew et al., 2012*) as potential co-binders of p53. Moreover, our analysis identified potential co-binders that to our knowledge have not been identified before, such as KDM1A, PRMT1, and GRHL2 for p63 and BRD9, ZNF131, and C17orf49 for p53. Importantly, these new potential co-binders appear to be unique to either p63 or p53, suggesting that they may contribute to shaping the DNA-binding landscapes that are specific to p63 and p53 (*Figure 5—figure supplement 4*).

## Identification of direct p63 target genes

Given that p63 regulates many target genes through enhancers (*Kouwenhoven et al., 2015a*; *Lin-Shiao et al., 2019*; *Lin-Shiao et al., 2018*; *Qu et al., 2018*; *Somerville et al., 2018*), straight forward integration of differential gene regulation data and p63 binding data based on proximity binding to a gene's TSS is unlikely to capture all direct p63 target genes. To resolve this issue, we integrated the p63 binding data and the *p63 Expression Score* based on enhancer:gene association information (*Fishilevich et al., 2017*) in addition to proximity binding to TSSs to predict direct p63 target genes. Given the large number of p63-binding sites identified (*Figure 5C and E*) and the high variance in p63-dependent gene regulation (*Figure 1B*), we employed conservative thresholds to identify high-probability target genes of p63. We only used p63-binding sites supported by at least half of the datasets ($\geq$10) that are linked through TSS proximity (within 5 kb) or double-elite enhancer:gene associations (*Fishilevich et al., 2017*) to genes with a |p63 Expression Score|| $\geq$ 8 (*Table 1* and *Figure 7—figure supplement 1*). Of note, many genes are associated with proximal and enhancer p63 binding, because many proximal promoters are also identified as double-elite enhancers in the database. The 180 (138 up-regulated and 42 down-regulated) genes that passed our conservative filtering contain many genes that are known direct p63 targets, such as *RAB38* (*Barton et al., 2010*), *S100A2* (*Kirschner et al., 2008*; *Lapi et al., 2006*), *HAS3* (*Compagnone et al., 2017*), *IRF6* (*Thomason et al., 2010*), *PTHLH* (*Somerville et al., 2018*), *GPX2* (*Yan and Chen, 2006*), *JAG1* (*Sasaki et al., 2002*), *MMP14* (*Lodillinsky et al., 2016*), *NRG1* (*Forster et al., 2014*), and *PLAC8* (*Gallant-Behm et al., 2012*). The identification of these well-established p63 target genes indicates the ability of our approach to identify bona fide candidates. Importantly, the integration of enhancer:gene associations enabled the identification of genes that are likely regulated by p63 through enhancers, such as *IL1B*, *MREG*, *MYO5A*, *RRP12*, *SNCA*, *AK4*, and *EHD4* (*Table 1* and *Figure 7—figure supplement 1*).

## A p63/SCC 28-gene set correlates with HNSC patient survival

Out of the 180 high-probability p63 target genes 32 (28 up- and four down-regulated) are also identified as being commonly up- or down-regulated in SCCs compared to non-SCC cancers (*Cancer Genome Atlas Research Network et al., 2018*; *Table 1*). Importantly, several of the genes commonly up-regulated by p63 as well as in SCC have been identified to promote SCC growth or invasion, such as *LAD1* (*Abe et al., 2019*), *TMEM40* (*Zhang et al., 2019*), *FGFBP1* (*Czubayko et al., 1997*), *IL1B* (*Lee et al., 2015*), *FAT2* (*Dang et al., 2016*), *FOSL1* (*Usui et al., 2012*), *LPAR3* (*Brusevold et al., 2014*), *MMP14* (*Pang et al., 2016*), and *RASSF6* (*Zheng et al., 2019a*). Therefore, we asked whether the set of 28 up-regulated direct p63 targets correlates with patient survival. To this end, we employed data of head and neck SCC (HNSC) patients from The Cancer Genome Atlas (TCGA). Notably, it is known that this cancer type frequently harbors amplified *TP63* (*Cancer Genome Atlas Network et al., 2015*). We find that expression levels of our gene set indeed correlate significantly negatively with HNSC patient survival (COX likelihood ratio test p=0.032). To determine whether expression levels of the set have an influence on the survival of HNSC patients, we subdivided the samples according to the average expression levels into four equally sized groups (low, low-med, med-high, high). While the sample group with low expression

**Table 1.** High-probability direct p63 target genes.

Genes identified as significantly up- or down-regulated in at least the half of all datasets (|p63 Expression Score| ≥ 8) that are linked to p63-binding sites supported by at least half of all datasets (≥10) through binding within 5 kb from their TSS or through double-elite enhancer-gene associations (*Fishilevich et al., 2017*). Using these thresholds we identified 138 and 42 high-probability candidates as directly up- and down-regulated by p63, respectively. Gene names marked in bold are also up- or down-regulated across SCCs (*Cancer Genome Atlas Research Network et al., 2018*).

| Gene symbol | p63 Expression Score | p63 binding within 5 kb from TSS | p63 binding linked through enhancer | Gene symbol | p63 Expression Score | p63 binding within 5 kb from TSS | p63 binding linked through enhancer |
|---|---|---|---|---|---|---|---|
| *DUSP6* | 14 | yes | yes | *FSCN1* | 8 | yes | yes |
| *RAB38* | 14 | yes | yes | *GINS3* | 8 | yes | no |
| *GSDME* | 13 | yes | yes | *GM2A* | 8 | yes | yes |
| **LAD1** | 13 | yes | yes | **HMGA2** | 8 | yes | yes |
| **S100A2** | 13 | yes | yes | *HSPA4L* | 8 | yes | yes |
| **TMEM40** | 13 | yes | yes | *JAG1* | 8 | yes | yes |
| **FGFBP1** | 12 | yes | yes | *KCTD12* | 8 | yes | no |
| *HAS3* | 12 | yes | no | *KIAA0930* | 8 | yes | yes |
| *NECTIN1* | 12 | yes | yes | *KIF14* | 8 | no | yes |
| *TCOF1* | 12 | yes | yes | *KIRREL1* | 8 | no | yes |
| *DUSP7* | 11 | yes | yes | *LIG1* | 8 | yes | yes |
| **IL1B** | 11 | no | yes | **LPAR3** | 8 | yes | yes |
| *MREG* | 11 | no | yes | *LRRFIP2* | 8 | no | yes |
| *PA2G4* | 11 | yes | no | *MALT1* | 8 | no | yes |
| *RGS20* | 11 | yes | no | *MAST4* | 8 | no | yes |
| *SDC1* | 11 | no | yes | *MCM3* | 8 | no | yes |
| **SFN** | 11 | yes | yes | **MMP14** | 8 | yes | yes |
| *STK17A* | 11 | yes | yes | *MMRN2* | 8 | yes | no |
| **VSNL1** | 11 | yes | yes | *NOM1* | 8 | yes | no |
| *ARHGAP25* | 10 | yes | yes | *NRCAM* | 8 | yes | yes |
| *CDCA4* | 10 | yes | yes | *NRG1* | 8 | no | yes |
| *DUSP11* | 10 | yes | no | *OAS3* | 8 | yes | yes |
| **FAT2** | 10 | yes | no | *PPFIBP1* | 8 | yes | yes |
| **FERMT1** | 10 | yes | yes | *PROCR* | 8 | yes | no |
| *IL4R* | 10 | yes | yes | *QSOX2* | 8 | yes | yes |
| *INPP1* | 10 | yes | yes | *RAD51C* | 8 | yes | yes |
| **IRF6** | 10 | no | yes | **RASSF6** | 8 | no | yes |
| *ITGA6* | 10 | no | yes | *RFX7* | 8 | yes | no |
| *KIZ* | 10 | yes | no | *SH3PXD2A* | 8 | no | yes |
| *MAPKBP1* | 10 | no | yes | *SLC1A5* | 8 | yes | yes |
| *MYO10* | 10 | yes | yes | *SLC2A9* | 8 | yes | yes |
| *MYO19* | 10 | yes | yes | *SLC37A2* | 8 | yes | no |
| *ORC1* | 10 | no | yes | *SMAD5* | 8 | yes | no |
| *PAK1* | 10 | yes | no | *SPATS2* | 8 | no | yes |
| **PTHLH** | 10 | yes | yes | *SSRP1* | 8 | no | yes |
| *SMTN* | 10 | yes | no | *TGFB1* | 8 | yes | yes |
| *WDFY2* | 10 | yes | no | *TMEM237* | 8 | yes | no |
| **XDH** | 10 | yes | yes | *TOMM34* | 8 | yes | no |

Table 1 continued

| Gene symbol | p63 Expression Score | p63 binding within 5 kb from TSS | p63 binding linked through enhancer | Gene symbol | p63 Expression Score | p63 binding within 5 kb from TSS | p63 binding linked through enhancer |
|---|---|---|---|---|---|---|---|
| ARHGDIB | 9 | yes | yes | TRIM7 | 8 | yes | yes |
| AURKB | 9 | yes | no | TRIP13 | 8 | yes | no |
| BTBD11 | 9 | yes | no | TSPAN5 | 8 | yes | no |
| C6orf106 | 9 | yes | no | TSR1 | 8 | no | yes |
| CARD10 | 9 | yes | yes | TYMS | 8 | yes | yes |
| CHAF1A | 9 | no | yes | UCK2 | 8 | yes | yes |
| CSTA | 9 | yes | no | UTP4 | 8 | no | yes |
| CYP27B1 | 9 | yes | no | YAP1 | 8 | yes | no |
| FEZ1 | 9 | yes | yes | YES1 | 8 | yes | yes |
| GNA15 | 9 | yes | no | ZFP36L2 | 8 | no | yes |
| GPX2 | 9 | yes | no | APH1B | -8 | no | yes |
| GSTP1 | 9 | yes | no | BIRC3 | -8 | yes | yes |
| HRAS | 9 | yes | yes | C9orf3 | -8 | yes | yes |
| IFI16 | 9 | yes | yes | CHST3 | -8 | no | yes |
| KREMEN1 | 9 | yes | yes | CPQ | -8 | no | yes |
| LDLR | 9 | yes | no | DUSP8 | -8 | yes | no |
| MAPK6 | 9 | yes | yes | EPCAM | -8 | no | yes |
| MYO5A | 9 | no | yes | ERBB2 | -8 | no | yes |
| NCAPH2 | 9 | yes | no | FBN1 | -8 | no | yes |
| NDE1 | 9 | yes | yes | ITFG1 | -8 | yes | no |
| NDST1 | 9 | yes | yes | LLGL2 | -8 | yes | yes |
| NIPAL4 | 9 | yes | yes | NCSTN | -8 | no | yes |
| PPIF | 9 | no | yes | OPN3 | -8 | no | yes |
| PPP4R4 | 9 | yes | no | PBX1 | -8 | yes | yes |
| PTTG1 | 9 | yes | yes | PDXK | -8 | no | yes |
| RAPGEF5 | 9 | yes | yes | PLAC8 | -8 | yes | yes |
| RNASE7 | 9 | yes | yes | S100A4 | -8 | no | yes |
| RRP12 | 9 | no | yes | SPOCK1 | -8 | no | yes |
| SERPINB13 | 9 | yes | no | TNS3 | -8 | no | yes |
| SNCA | 9 | no | yes | ARL6IP5 | -9 | no | yes |
| STX6 | 9 | yes | no | COBL | -9 | no | yes |
| AK4 | 8 | no | yes | CUEDC1 | -9 | yes | yes |
| ARHGAP23 | 8 | yes | yes | GSN | -9 | yes | no |
| ASCC3 | 8 | yes | yes | PDGFC | -9 | yes | yes |
| BRCA1 | 8 | yes | no | PGPEP1 | -9 | no | yes |
| BTBD10 | 8 | yes | yes | PLXNB2 | -9 | yes | yes |
| CCNK | 8 | yes | no | PXDN | -9 | no | yes |
| CCT4 | 8 | yes | no | RALGPS1 | -9 | yes | yes |
| CD44 | 8 | yes | yes | ROR1 | -9 | yes | no |
| CDC42SE1 | 8 | yes | no | SLC16A5 | -9 | yes | yes |
| CDCA7 | 8 | yes | no | TM4SF1 | -9 | yes | yes |
| COL17A1 | 8 | yes | no | ALDH3B1 | −10 | yes | yes |

Table 1 continued on next page

*Table 1 continued*

| Gene symbol | p63 Expression Score | p63 binding within 5 kb from TSS | p63 binding linked through enhancer | Gene symbol | p63 Expression Score | p63 binding within 5 kb from TSS | p63 binding linked through enhancer |
|---|---|---|---|---|---|---|---|
| CRKL | 8 | yes | yes | *CYP1B1* | −10 | no | yes |
| DRAP1 | 8 | yes | yes | HHAT | −10 | yes | yes |
| EHD4 | 8 | no | yes | MEGF8 | −10 | no | yes |
| ERCC6L | 8 | no | yes | PTGES | −10 | yes | no |
| ESRP1 | 8 | no | yes | PTTG1IP | −10 | no | yes |
| *FABP5* | 8 | yes | no | RPS27L | −10 | yes | yes |
| FANCI | 8 | yes | yes | SECTM1 | −10 | yes | yes |
| FLOT2 | 8 | yes | no | SLC22A5 | −10 | yes | no |
| *FOSL1* | 8 | yes | yes | TNFSF15 | −10 | yes | yes |
| FRMD4B | 8 | yes | no | SRD5A3 | −11 | yes | no |

had the most favorable prognosis, the null hypothesis could not be rejected in the direct comparison with patients with high average expression levels (p=0.090; *Figure 8A*). However, upon contrasting the low-expression group with all remaining samples, a significant improvement of survival was detected (p=0.024; *Figure 8B*). Expression of the 28-gene set correlated positively with p63 expression when p63 expression was rather low (FPKM <20), but showed a saturation and no further correlation when p63 expression was high (FPKM >20; *Figure 8C and D*, and *Figure 8—figure supplement 1*). This indicates that p63 levels influence the 28-gene set in a switch-like manner where a saturation of p63-dependent activation is quickly reached in HNSC cells. Together, these findings indicate that the genes commonly up-regulated by p63 and in SCC influence the prognosis of HNSC patients. Taken together, this finding calls for a more detailed assessment of ubiquitous p63/SCC genes as biomarkers in the future.

## Discussion

Although p63 (ΔNp63) is known as master regulator in epidermis development and more recently emerged as a key oncogenic factor in SCC, a comprehensive assessment of the GRN commonly controlled by p63 and its comparison to the GRN commonly controlled by the closely related tumor suppressor p53 has been missing. An increasing number of available high-throughput datasets enabled us to generate ranked lists of p63-regulated genes and p63-bound DNA sites that together reveal high-probability direct p63 target genes regulated by p63 across cells of multiple origins. Because p63 target genes, very much like p53 target genes (*Fischer, 2020*; *Fischer, 2019*), differ substantially between mouse and human (*Sethi et al., 2017*), many p63 target genes initially described in mouse could not be confirmed to be p63-regulated in this study using human data. Given that p63-binding sites are frequently associated with enhancer regions and enhancer identity, we have integrated enhancer:gene associations to identify target genes that are regulated by p63 through direct binding to associated enhancers. This approach enabled the identification of novel direct target genes that are missed by standard analyses that employ only TSS proximity (*Table 1* and *Figure 7—figure supplement 1*).

Given the similarity between their DBDs, it has been a long-standing question how p53 and p63 bind to distinct sites in the genome and how these sites differ from another. Several studies found differences in the biochemical properties of p53 and p63 that could affect their DNA binding specificity (*Enthart et al., 2016*; *Lokshin et al., 2007*; *Sauer et al., 2008*; *Tichý et al., 2013*). Various studies aimed to identify the precise p63 recognition motif and its difference from the p53RE using either SELEX (*Ortt and Sinha, 2006*; *Perez et al., 2007*) or ChIP-seq data (*Kouwenhoven et al., 2010*; *McDade et al., 2014*; *Yang et al., 2006*), yet these studies reported different features as being unique for p63 compared to p53 DNA recognition. By combining multiple ChIP-seq datasets we have contributed here to better distinguish between sites commonly bound by p53 and p63

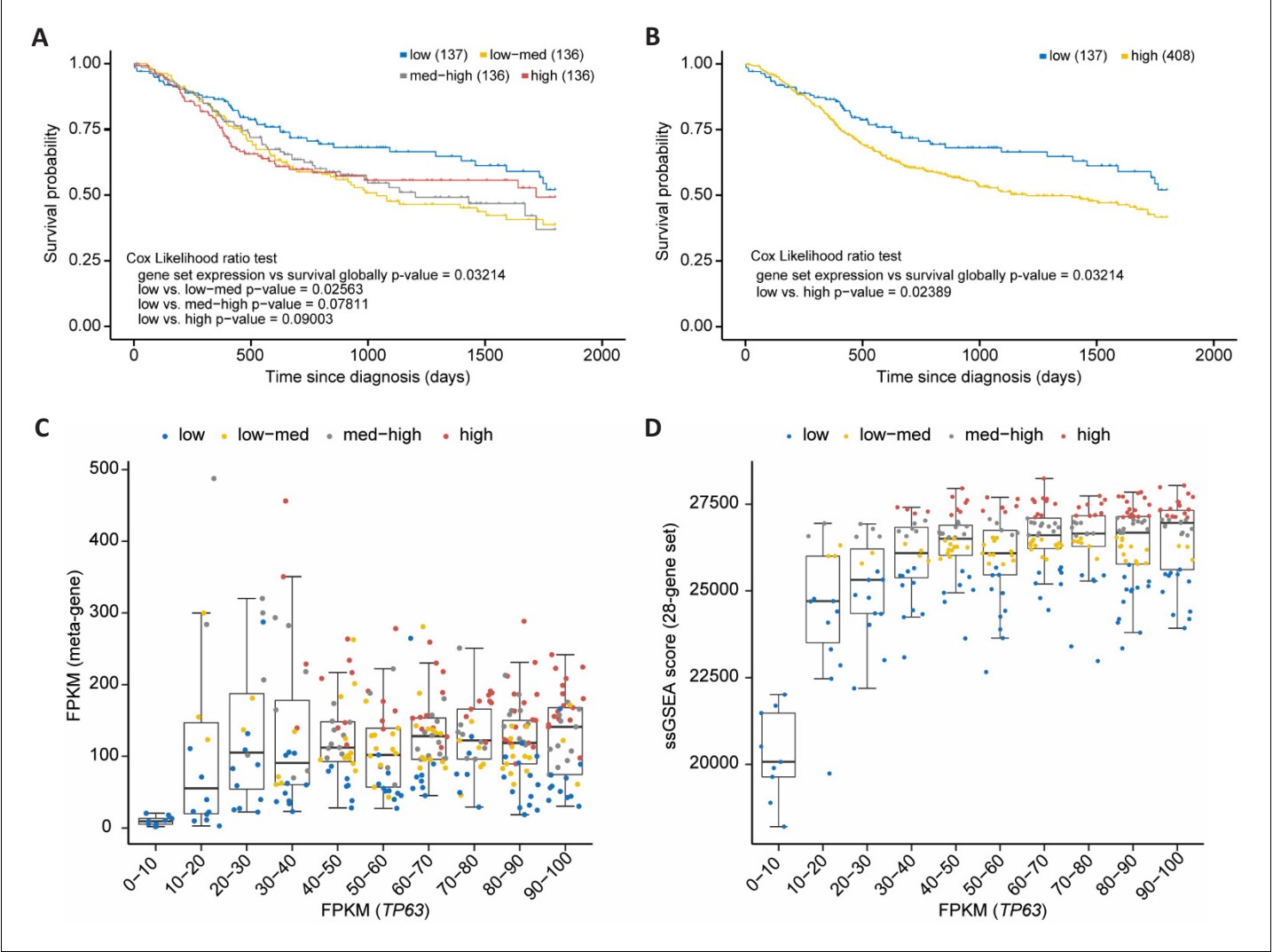

**Figure 8.** p63/SCC 28-gene set correlates with poorer survival in HNSC. Kaplan-Meier plots of TCGA HNSC patient survival data. (**A**) Patients were subdivided in four equally sized subgroups based on expression levels of the 28-gene set. The results suggest a poorer survival of patients with an up-regulated expression of the set genes. (**B**) To corroborate this finding patients of the subgroups low-med, med-high, and high from (**A**) were joined to form a new high group. Boxplot in bins of 10 of *TP63* FPKM expression values in TCGA HNCS patient sample data compared to (**C**) FPKM values of a meta-gene comprising the 28-gene set and (**D**) ssGSEA scores of the 28-gene set. X-axis is right-censored at 100 to better visualize the effect. The full graph is displayed in *Figure 8—figure supplement 1*.

The online version of this article includes the following figure supplement(s) for figure 8:

**Figure supplement 1.** Extension of *Figure 8C and D*.

across cell types and sites that are unique to p53 or p63 (*Figure 5E*). Most importantly, our results could explain why a substantial fraction of DNA sites is occupied exclusively by p53 or p63. While most sites bound by p53 are also commonly occupied by p63 (*Figure 5E* and *Figure 5—figure supplement 1A*), single half-sites and half-sites separated by spacers underlie many sites that are only bound by p53 (*Figures 5F* and *6*), supporting earlier findings whereby p53 can be recruited through spacer-containing motifs (*Vyas et al., 2017*). However, while spacers reportedly have been identified in fifty percent of 200 analyzed p53REs (*Vyas et al., 2017*), our genome-wide quantification of motifs underlying 7705 high confidence p53 peaks based on an unbiased motif search using HOMER revealed that only 1.1% to 5.1% of the p53 peaks contain p53REs with 1 bp spacers or half sites that are possibly separated by longer spacers (*Figure 6*). Mechanistically, our results imply that relying on the CWWG core motif and the flanking regions may enable p53 to bind to those sites. In

contrast, the two CNNG core motifs that underlie p63, but not p53REs, offer an explanation why a substantial fraction of DNA sites is bound exclusively by p63 (*Figures 5F* and *7*), supporting one of the models established earlier (*McDade et al., 2014*). Notably, p63's ability to bind to a greater variety of recognition motifs likely underlies the markedly greater number of p63 compared to p53-binding sites in the genome. In addition, our motif search indicates that factors bound to AP-1 (bZIP) and bHLH motifs may specifically support p63 binding (*Figure 5—figure supplement 3*), and transcription factor enrichment analysis identified the bZIP TF MAF, the TF GRHL2, the chromatin remodeler BANF1, the histone methyltransferase PRMT1, and the ZNF750/KDM1A/KLF4 complex, which was previously shown to operate downstream of p63 (*Boxer et al., 2014*), as potential co-binders that could help to facilitate p63 binding to certain genomic loci (*Figure 5—figure supplement 4*). Considering its pioneer role, p63 could vice versa enable the binding of these TFs to the respective loci. Given that p63 and p73 form stable heterotetramers (*Gebel et al., 2016*), p73 may possess binding specificities that are highly similar to those identified for p63. Our results indicate that our approach could serve as a blueprint to distinguish DNA recognition motifs, binding sites, co-factors, and target genes of TF siblings more precisely. Our iterative de novo search algorithm enabled the identification of spacer-containing p53REs, indicating that our approach uncovers second-tier TF binding motifs invisible to standard approaches. Moreover, the results provide insights to the p63 DNA binding repertoire in unprecedented depth (*Figure 5F*).

Consistent with results from an earlier genome-wide study (*Yang et al., 2006*), our findings imply that p63 is more frequently involved in a direct up-regulation as opposed to a direct down-regulation of target genes (*Figure 3A* and *Figure 7—figure supplement 1*). Mechanistically, p63 has been shown to up-regulate target genes through its alternative TAD located at the N-terminus while the C-terminus is important for down-regulation (*Helton et al., 2006*). Exogenous expression of different isoforms of p53 family members and their antagonistic effects on target gene promoters in luciferase reporter assays suggested a model whereby p63 exhibits a dominant negative effect on other p53 family members (*Mundt et al., 2010*; *Westfall et al., 2003*; *Yang et al., 1998*). Inconsistent with its reputation as dominant negative regulator of p53, however, genome-wide studies showed that the groups of p63-regulated genes and p53-regulated genes show only very little overlap (*Gallant-Behm et al., 2012*). A recent analysis of DNA sites bound and of genes regulated by p53 and p63 revealed that p63 is more likely to support than to inhibit p53 activity (*Karsli Uzunbas et al., 2019*). Our analysis further supports the notion that p63 does not commonly interfere with target gene up-regulation by p53 but that except for cell cycle genes they regulate largely distinct gene sets (*Figure 4*).

We identify several candidate TFs that may operate downstream of p63 and that may serve as transitional nodes in the p63 GRN. In addition to known mediators of p63-dependent gene regulation, such as MYC and KLF4, we identify AR and its co-factor ZMIZ1, SP1, FLI1, and NANOG as novel candidate nodes in the p63 GRN (*Figure 3A*). In agreement with the tumor suppressor role of p53 and the oncogenic role of p63, we find that cell cycle genes are antagonistically regulated by p53 and p63 (*Figures 2A* and *4A*). On the one hand, cell cycle genes are well-known to be down-regulated by p53 indirectly through the cyclin-dependent kinase inhibitor p21 and the cell cycle repressor complexes DREAM and RB-E2F (*Fischer et al., 2016a*; *Fischer et al., 2016b*; *Schade et al., 2019*; *Uxa et al., 2019*). On the other hand, cell cycle genes are down-regulated upon loss of p63 and this p63-dependent regulation reportedly occurs through regulating p21 signaling and the DREAM component p130 (*McDade et al., 2011*; *Truong et al., 2006*). In addition to indirect effects, we also predicted multiple cell cycle genes as direct p63 targets (*Table 1*). Consequently, a loss of p63 may substantially contribute to the effect of p53 in reducing cell cycle gene expression (*Figure 3D*). In addition of p63's role in driving the expression of some cell cycle genes, the entire set of cell cycle genes may be subsequently up-regulated indirectly through p63's pro-proliferative targets. While the up-regulation of cell cycle genes occurs in most cancers (*Whitfield et al., 2006*), we find that p63 additionally regulates genes that are specifically altered across SCCs (*Figure 2D*). These results underscore the critical role of p63 and its target genes in determining the transcriptional profile of SCC. An example of a p63 target in SCC is NRG1, which can be inhibited to block SCC proliferation and tumor growth (*Hegde et al., 2019*). The resource of genes commonly regulated by p63 provided here may help to identify targets that can be exploited therapeutically. We provided a showcase example, where expression levels of the 28 p63 target genes that are commonly up-regulated by p63 and in SCC (*Table 1*) correlate significantly with poorer survival of HNSC patients (*Figure 8*).

Thus, this 28-gene set may contain particularly promising candidates for therapeutic interventions and for the use as biomarkers.

## Materials and methods

### Re-analysis and integration of publicly available gene expression profiling datasets

We re-analyzed publicly available p63-dependent gene expression profiling datasets. As a first quality requirement, we only included datasets for re-analysis that contained at least two biological replicates for the treatment as well as for the control condition. All microarray datasets were available at a pre-processed stage at the Gene Expression Omnibus (GEO) and we re-analyzed these datasets with GEO2R to obtain fold expression changes and Benjamini Hochberg-corrected p-values (*Clough and Barrett, 2016*). Gene identifiers were mapped to Ensembl Gene IDs using the Ensembl annotation data (*Cunningham et al., 2019*). All RNA-seq datasets have been retrieved through GEO from the Sequence Read Archive (SRA) (*Leinonen et al., 2011*). We employed our RNA-seq analysis pipeline to obtain fold expression changes and p-values adjusted for multiple testing. Briefly, we utilized Trimmomatic (*Bolger et al., 2014*) v0.39 (5nt sliding window approach, mean quality cutoff 22) for read quality trimming according to inspections made from FastQC (https://www.bioinformatics. babraham.ac.uk/projects/fastqc/) v0.11.8 reports. Clipping was performed using Cutadapt v2.3 (*Martin, 2011*). Potential sequencing errors were detected and corrected using Rcorrector v1.0.3.1 (*Song and Florea, 2015*). Ribosomal RNA (rRNA) transcripts were artificially depleted by read alignment against rRNA databases through SortMeRNA v2.1 (*Kopylova et al., 2012*). The preprocessed data was aligned to the reference genome hg38, retrieved along with its gene annotation from Ensembl v.92 (*Cunningham et al., 2019*). For read alignment, we used the splice-aware mapping software segemehl (*Hoffmann et al., 2014*; *Hoffmann et al., 2009*) v0.3.4 with adjusted accuracy (95%). Mappings were filtered by Samtools v1.9 (*Li et al., 2009*) for uniqueness and properly aligned mate pairs. Read quantification was performed on exon level using featureCounts v1.6.5 (*Liao et al., 2014*), parametrized according to the strand specificity inferred through RSeQC v3.0.0 (*Wang et al., 2012*). Differential gene expression and its statistical significance was identified using DESeq2 v1.20.0 (*Love et al., 2014*). Information on the samples that were compared for each dataset is included in *Supplementary file 1*. Given that all RNA-seq data was derived from PolyA-enriched samples, we only included Ensembl transcript types 'protein_coding', 'antisense', 'lincRNA' and 'TEC' in our analysis. Common thresholds for adj. p-value$\leq$0.05 were applied.

### Generation of the p63 expression score

For 19,156 genes covered by at least three datasets including a minimum of one RNA-seq dataset, a *p63 Expression Score* was calculated as the number of datasets that find the gene to be significantly up-regulated minus the number of datasets that find the gene to be significantly down-regulated in dependence on p63. This meta-analysis resulted in 27 *p63 Expression Score* gene groups because no gene was identified as up-regulated in all 16 or 15 datasets or down-regulated in all 16, 15, 14, or 13 datasets.

### Enrichment analyses

Gene set enrichment analysis (GSEA) was performed using GSEA (http://software.broadinstitute. org/gsea/) with 'H', 'C2', and 'C6' gene sets from MSigDB v7.0 (*Subramanian et al., 2005*) and custom panSCC gene sets derived from Table S1C in *Cancer Genome Atlas Research Network et al., 2018*. GSEA was performed on a pre-ranked list of genes that were ranked primarily by *p63 Expression Score* and secondarily by median log$_2$(fold-change) to obtain unique ranks.

Enrichment of transcription factor binding to genes with high ($\geq$8) or low ($\leq -8$) *p63 Expression Score* was identified using the results section 'ENCODE and ChEA Consensus TFs from ChIP-X' from Enrichr (*Kuleshov et al., 2016*).

### Integration of publicly available p63 and p53 binding data

Peak datasets from p63 ChIP-seq experiments were retrieved from CistromeDB (*Zheng et al., 2019b*; *Supplementary file 1*). When replicate experiments were available, all peaks were used that

have been identified in at least two replicates. A similar collection of p53 peak datasets has been described previously (*Fischer, 2019*). To intersect multiple peak files Bedtools 'multiinter' was used and to identify overlapping and non-overlapping peaks Bedtools 'intersect' was employed (*Quinlan and Hall, 2010*).

## Motif search

Known p53 and p63REs were identified using the 'known motifs' in HOMER v4.10 with default options and *-size given* (*Heinz et al., 2010*). De novo motif discovery was performed with options *-size given -len 10,15,20,25 -mis 5 S 10*.

## Identification of potential co-factors

We used the CistromeDB toolkit (*Zheng et al., 2019b*) to identify TFs that display ChIP-seq peaksets that are significantly similar to our 'unique p53', 'unique p63', and 'p53+p63' peaksets.

## Survival and expression correlation analysis

For the 28-gene set, single-sample enrichment scores were derived from FPKM normalized gene expression values of 546 HNSC patient samples. To this end, we utilized the official GenePattern single sample gene set enrichment analysis (ssGSEA) codebase v10.0.3 (*Barbie et al., 2009*; https://github.com/GSEA-MSigDB/ssGSEA-gpmodule). A sample score represents the coordinately up- or down-regulated expression of all genes within one set as its signature (*Barbie et al., 2009*). Kaplan-Meier plots and correlation analyses were performed on TCGA time to event and event occurrence information using the R survival package v3.2–3. Following the TCGA standard for HNSC (*Cancer Genome Atlas Network et al., 2015*), survival analyses were right-censored at 60 months (1800 days) to avoid non-cancer-related events. The Cox proportional hazards model was used to investigate the association of patient survival time and the combined expression levels of the 28-gene set. Subsequently, we subdivided the expression scores into four equally sized categorical groups (high, med-high, med-low, low). The rates of occurrence of events over time were compared between these groups using the fitted COX PH model.

We retrieved read quantification data 'HTSeq - Counts' from 546 samples of the TCGA project HNSC utilizing the R package TCGAbiolinks v2.18.0 (*Colaprico et al., 2016*). Per sample, all read counts of the 28-gene set were merged into an artificially created metagene. Subsequently, we calculated normalized expression values per gene as fragments per kilobase million, where the length of a gene corresponds to the lengths of its exons assigned to either the canonical transcript (CCDS) or the longest transcript according to hg38 Ensembl annotation v92. *TP63* FPKM values were plotted against the meta-gene FPKM value or the ssGSEA derived gene set scores (see above).

# Additional information

## Funding

| Funder | Grant reference number | Author |
| --- | --- | --- |
| Deutsche Forschungsgemeinschaft | FI 1993/2-1 | Martin Fischer |
| Bundesministerium für Bildung und Forschung | 031L016D | Steve Hoffmann |
| Leibniz Association | Open Access Fund | Martin Fischer Steve Hoffmann |

The funders had no role in study design, data collection and interpretation, or the decision to submit the work for publication.

## Author contributions

Konstantin Riege, Data curation, Software, Formal analysis, Investigation, Visualization, Methodology, Writing - review and editing; Helene Kretzmer, Formal analysis, Methodology, Writing - review and editing; Arne Sahm, Methodology, Writing - review and editing; Simon S McDade, Investigation,

Writing - review and editing; Steve Hoffmann, Resources, Supervision, Funding acquisition, Investigation, Writing - review and editing; Martin Fischer, Conceptualization, Data curation, Formal analysis, Supervision, Funding acquisition, Validation, Investigation, Visualization, Methodology, Writing - original draft, Project administration, Writing - review and editing

#### Author ORCIDs
Arne Sahm http://orcid.org/0000-0002-7330-1790
Simon S McDade https://orcid.org/0000-0002-3024-4773
Martin Fischer https://orcid.org/0000-0002-3429-1876

#### Decision letter and Author response
Decision letter https://doi.org/10.7554/eLife.63266.sa1
Author response https://doi.org/10.7554/eLife.63266.sa2

## Additional files
#### Supplementary files
• Supplementary file 1. Detailed information on publicly available p63-dependent gene expression profiling and p63 ChIP-seq datasets that were integrated in this study.

• Supplementary file 2. Meta-analysis from 16 p63-dependent gene expression information datasets (listed in Suppelemtary File 1) to generate the *p63 Expression Score* for 19,156 human genes.

• Supplementary file 3. p63- and p53-binding sites identified in at least 5 out of 20 and 28 ChIP-seq datasets, respectively. Binding sites are listed with their ChIP-seq dataset support and highest scoring p63 response elements (p63REs) or p53REs. Genes associated with p63-binding sites through proximal TSS binding or enhancers are listed.

• Transparent reporting form

#### Data availability
All data generated or analysed during this study are included in the manuscript and supporting files.

The following previously published datasets were used:

| Author(s) | Year | Dataset title | Dataset URL | Database and Identifier |
|---|---|---|---|---|
| Bao X, Rubin AJ, Qu K, Zhang J, Giresi PG, Chang HY, Khavari PA | 2015 | A novel ATAC-seq approach reveals lineage-specific reinforcement of the open chromatin landscape via cooperation between BAF and p63 | https://www.ncbi.nlm.nih.gov/geo/query/acc.cgi?acc=GSE67382 | NCBI Gene Expression Omnibus, GSE67382 |
| Kouwenhoven EN, van Heeringen SJ, Tena JJ, Oti M, Dutilh BE, Alonso ME, de la Calle-Mustienes E, Smeenk L, Rinne T, Parsaulian L, Bolat E, Jurgelenaite R, Huynen MA, Hoischen A, Veltman JA, Brunner HG, Roscioli T, Oates E, Wilson M, Manzanares M, Gómez-Skarmeta JL, Stunnenberg HG, Lohrum M, van Bokhoven H, Zhou H | 2010 | Genome-Wide Profiling of p63 DNA-Binding Sites Identifies an Element that Regulates Gene Expression during Limb Development in the 7q21 SHFM1 Locus | https://www.ncbi.nlm.nih.gov/geo/query/acc.cgi?acc=GSE17611 | NCBI Gene Expression Omnibus, GSE17611 |
| Kouwenhoven EN, Oti M, Niehues H, | 2015 | Transcription factor p63 bookmarks and regulates dynamic enhancers | https://www.ncbi.nlm.nih.gov/geo/query/acc. | NCBI Gene Expression Omnibus, |

| | | | | |
|---|---|---|---|---|
| van Heeringen SJ, Schalkwijk J, Stunnenberg HG, Bokhoven H, Zhou H | | during epidermal differentiation | cgi?acc=GSE59824 | GSE59824 |
| McDade SS, Henry AE, Pivato GP, Kozarewa I, Mitsopoulos C, Fenwick K, Assiotis I, Hakas J, Zvelebil M, Orr N, Lord CJ, Patel D, Ashworth A, McCance DJ | 2012 | Genome-wide analysis of p63 binding sites identifies AP-2 factors as co-regulators of epidermal differentiation | https://www.ncbi.nlm. nih.gov/geo/query/acc. cgi?acc=GSE32061 | NCBI Gene Expression Omnibus, GSE32061 |
| McDade SS, Patel D, Moran M, Campbell J, Fenwick K, Kozarewa I, Orr NJ, Lord CJ, Ashworth AA, McCance DJ | 2014 | Genome-wide characterization reveals complex interplay between TP53 and TP63 in response to genotoxic stress | https://www.ncbi.nlm. nih.gov/geo/query/acc. cgi?acc=GSE56674 | NCBI Gene Expression Omnibus, GSE56674 |
| Olsen JR, Oyan AM, Rostad K, Hellem MR, Liu J, Li L, Micklem DR, Haugen H, Lorens JB, Rotter V, Ke XS, Lin B, Kalland KH | 2013 | p63 attenuates epithelial to mesenchymal potential in an experimental prostate cell model | https://www.ncbi.nlm. nih.gov/geo/query/acc. cgi?acc=GSE43111 | NCBI Gene Expression Omnibus, GSE43111 |
| Saladi SV, Ross K, Karaayvaz M, Tata PR, Mou H, Rajagopal J, Ramaswamy S, Ellisen LW | 2017 | ACTL6A Is Co-Amplified with p63 in Squamous Cell Carcinoma to Drive YAP Activation, Regenerative Proliferation, and Poor Prognosis | https://www.ncbi.nlm. nih.gov/geo/query/acc. cgi?acc=GSE88859 | NCBI Gene Expression Omnibus, GSE88859 |
| Vasilaki E, Morikawa M, Koinuma D, Mizutani A, Hirano Y, Ehata S, Sundqvist A, Kawasaki N, Cedervall J, Olsson AK, Aburatani H, Moustakas A, Miyazono K, Heldin CH | 2016 | Ras and TGF-$\beta$ signaling enhance cancer progression by promoting the $\Delta$Np63 transcriptional program | https://www.ncbi.nlm. nih.gov/geo/query/acc. cgi?acc=GSE60814 | NCBI Gene Expression Omnibus, GSE60814 |
| Watanabe H, Ma Q, Peng S, Adelmant G, Swain D, Song W, Fox C, Francis JM, Pedamallu CS, DeLuca DS, Brooks AN, Wang S, Que J, Rustgi AK, Wong K, Ligon KL, Liu XS, Marto JA, Meyerson M, Bass AJ | 2014 | SOX2 and p63 colocalize at genetic loci in squamous cell carcinomas | https://www.ncbi.nlm. nih.gov/geo/query/acc. cgi?acc=GSE46837 | NCBI Gene Expression Omnibus, GSE46837 |
| Zarnegar BJ, Webster DE, Lopez-Pajares V, Vander Stoep Hunt B, Qu K, Yan KJ, Berk DR, Sen GL, Khavari PA | 2012 | Genomic profiling of a human organotypic model of AEC syndrome reveals ZNF750 as an essential downstream target of mutant TP63 | https://www.ncbi.nlm. nih.gov/geo/query/acc. cgi?acc=GSE33571 | NCBI Gene Expression Omnibus, GSE33571 |
| Abraham CG, Ludwig MP, Andrysik Z, Pandey A, Joshi M, Galbraith MD, Sullivan KD, Espinosa JM | 2018 | $\Delta$Np63$\alpha$ Suppresses TGFB2 Expression and RHOA Activity to Drive Cell Proliferation in Squamous Cell Carcinomas | https://www.ncbi.nlm. nih.gov/geo/query/acc. cgi?acc=GSE111619 | NCBI Gene Expression Omnibus, GSE111619 |
| Abraham CG, Ludwig MP, Andrysik Z, Pandey A, Joshi M, | 2006 | p63 regulates an adhesion programme and cell survival in epithelial cells | https://www.ncbi.nlm. nih.gov/geo/query/acc. cgi?acc=GSE20286 | NCBI Gene Expression Omnibus, GSE20286 |

| | | | | |
|---|---|---|---|---|
| Galbraith MD, Sullivan KD, Espinosa JM | | | | |
| Gallant-Behm CL, Ramsey MR, Bensard CL, Nojek I, Tran J, Liu M, Ellisen LW, Espinosa JM | 2012 | ΔNp63α represses anti-proliferative genes via H2A.Z deposition | https://www.ncbi.nlm.nih.gov/geo/query/acc.cgi?acc=GSE40462 | NCBI Gene Expression Omnibus, GSE40462 |
| Karsli Uzunbas G, Ahmed F, Sammons MA | 2019 | Control of p53-dependent transcription and enhancer activity by the p53 family member p63 | https://www.ncbi.nlm.nih.gov/geo/query/acc.cgi?acc=GSE111009 | NCBI Gene Expression Omnibus, GSE111009 |
| Lin-Shiao E, Lan Y, Welzenbach J, Alexander KA, Zhang Z, Knapp M, Mangold E, Sammons M, Ludwig KU, Berger SL | 2019 | p63 establishes epithelial enhancers at critical craniofacial development genes | https://www.ncbi.nlm.nih.gov/geo/query/acc.cgi?acc=GSE126397 | NCBI Gene Expression Omnibus, GSE126397 |
| Saladi SV, Ross K, Karaayvaz M, Tata PR, Mou H, Rajagopal J, Ramaswamy S, Ellisen LW | 2017 | ACTL6A Is Co-Amplified with p63 in Squamous Cell Carcinoma to Drive YAP Activation, Regenerative Proliferation, and Poor Prognosis | https://www.ncbi.nlm.nih.gov/geo/query/acc.cgi?acc=GSE88833 | NCBI Gene Expression Omnibus, GSE88833 |
| Saladi SV, Ross K, Karaayvaz M, Tata PR, Mou H, Rajagopal J, Ramaswamy S, Ellisen LW | 2017 | ACTL6A Is Co-Amplified with p63 in Squamous Cell Carcinoma to Drive YAP Activation, Regenerative Proliferation, and Poor Prognosis | https://www.ncbi.nlm.nih.gov/geo/query/acc.cgi?acc=GSE88832 | NCBI Gene Expression Omnibus, GSE88832 |
| Somerville TDD, Xu Y, Miyabayashi K, Tiriac H, Cleary CR, Maia-Silva D, Milazzo JP, Tuveson DA, Vakoc CR | 2018 | TP63-Mediated Enhancer Reprogramming Drives the Squamous Subtype of Pancreatic Ductal Adenocarcinoma | https://www.ncbi.nlm.nih.gov/geo/query/acc.cgi?acc=GSE115462 | NCBI Gene Expression Omnibus, GSE115462 |
| Watanabe H, Ma Q, Peng S, Adelmant G, Swain D, Song W, Fox C, Francis JM, Pedamallu CS, DeLuca DS, Brooks AN, Wang S, Que J, Rustgi AK, Wong K, Ligon KL, Liu XS, Marto JA, Meyerson M, Bass AJ | 2014 | SOX2 and p63 colocalize at genetic loci in squamous cell carcinomas | https://www.ncbi.nlm.nih.gov/geo/query/acc.cgi?acc=GSE47058 | NCBI Gene Expression Omnibus, GSE47058 |
| Wu N, Rollin J, Masse I, Lamartine J, Gidrol X | 2012 | p63 regulates human keratinocyte proliferation via MYC-regulated gene network and differentiation commitment through cell adhesion-related gene network | https://www.ncbi.nlm.nih.gov/geo/query/acc.cgi?acc=GSE17394 | NCBI Gene Expression Omnibus, GSE17394 |
| Zarnegar BJ, Webster DE, Lopez-Pajares V, Vander Stoep Hunt B, Qu K, Yan KJ, Berk DR, Sen GL, Khavari PA | 2012 | Genomic profiling of a human organotypic model of AEC syndrome reveals ZNF750 as an essential downstream target of mutant TP63 | https://www.ncbi.nlm.nih.gov/geo/query/acc.cgi?acc=GSE33495 | NCBI Gene Expression Omnibus, GSE33495 |

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
