## [Decision Letter]

**Acceptance summary:**

This manuscript provides a comprehensive analysis of p53 and p63 targets, identified through a variety of approaches in several studies and in different experimental contexts. The authors have achieved a daunting task by collecting all the ChIP and gene expression data to constitute the first database of p53/p63 response element and thus perform a meta-analysis. It is a much necessary work as such a database paves the way to break the p53/p63 code and thus better predict pathologies and the response to treatment.

**Decision letter after peer review:**

Thank you for submitting your article entitled "Dissecting the DNA binding landscape and gene regulatory network of p63 and p53" for consideration by *eLife*. Your revised article has been reviewed by three peer reviewers, one of whom is a member of our Board of Reviewing Editors, and the evaluation has been overseen by Maureen Murphy as the Senior Editor. There are a few issues that the reviewers feel need to be addressed in the manuscript. These are outlined below.

Summary:

This manuscript provides a comprehensive analysis of p53 and p63 targets, identified through a variety of approaches in several studies and in different experimental contexts. This is a very important work. The authors have achieved a daunting task by collecting all the ChIP and gene expression data to constitute the first database of p53/p63 response element and thus perform a meta-analysis. It is a much necessary work as such a database paves the way to break the p53/p63 code and thus better predict pathologies and the response to treatment (particularly when polymorphisms, somatic mutations or DNA methylation occur in the response elements). Statistical analysis sounds adequate and well designed. The data supports the conclusions. Addressing the following comments will further improve the study before publication in *eLife*.

Essential revisions:

1) "Notably, when DREAM targets were already down-regulated through Nutlin treatment in MCF10A cells, loss of p63 decreased their expression even further (Figure 3C) (Karsli Uzunbas et al., 2019), which indicates a possible cumulative effect that is independent of p53 regulatory functions as has been suggested previously (McDade et al., 2014)." This concept is not convincing. In Figure 3C you indeed show shp63 and shp63/Nutlin conditions. In the text you mention the effect of only Nutlin, which is not shown in the graph. Please explain. Moreover, why do you hypothesize "a possible cumulative effect that is independent of p53 regulatory functions"? MCF10A should have TP53 wild type. So, why excluding an effect of Nutlin on p53?

2) Figure 9: Is the identified p63-signature differentially expressed in reference to the state of p63 (over expressed and/or amplified) in publicly available datasets of HNSC with expression and genomic data? Also, in Figure 9, is the identified signature prognostic specifically in HNSC and not in other SCC types?

3) To increase the long-term usability of this work for future studies, can the authors report the gene and the DNA sequence of the response element in the Table 3 as the reported position may change with the updating of the human genome sequence (GRCh38)?

4) The authors report that there are many more number of p63 binding sites in the genome as compared to the number of p53 binding sites. Is this because basal p63 levels are much higher than basal p53 levels? Can the authors comment on the relative molecules per cell of p63 and p53? Or is it possible that this difference is because p63 requires shorter sequences in the DNA and is assisted by a number of cofactors?

---

## [Author Response]

Essential revisions:1) "Notably, when DREAM targets were already down-regulated through Nutlin treatment in MCF10A cells, loss of p63 decreased their expression even further (Figure 3C) (Karsli Uzunbas et al., 2019), which indicates a possible cumulative effect that is independent of p53 regulatory functions as has been suggested previously (McDade et al., 2014)." This concept is not convincing. In Figure 3C you indeed show shp63 and shp63/Nutlin conditions. In the text you mention the effect of only Nutlin, which is not shown in the graph. Please explain. Moreover, why do you hypothesize "a possible cumulative effect that is independent of p53 regulatory functions"? MCF10A should have TP53 wild type. So, why excluding an effect of Nutlin on p53?

We thank the reviewers for the remark, and we take this opportunity to add a visualization and to improve the description of our results and their underlying data. As correctly noted by the reviewer, MCF10A cells possess functional wild-type p53 in which DREAM target genes are down-regulated following p53 induction by Nutlin treatment. The difference displayed in shControl versus shp63 in Nutlin-treated MCF10A cells indicates that although p53 is active in these cells leading to the activation of DREAM and repression of its targets, we find that these genes are further down-regulated upon depletion of p63. To our understanding, this is indicative of a cumulative effect of p53-dependent DREAM activation and a loss of p63 activator activity at DREAM targets, largely affecting cell cycle genes. Following the reviewers’ comment, we added new Figure 3D that better visualizes this effect and improved the description in the respective Results paragraph:

“Notably, most datasets on p63-dependent gene expression were derived from cells in which p63 was overexpressed or depleted, without additional treatments. […] Strikingly, depletion of p63 decreased their expression even further without affecting *CDKN1A* (p21) levels (Figure 3D), which indicates a possible cumulative effect that is independent of p53 regulatory functions.”

Besides, we extended the discussion on this phenomenon in the Discussion section:

“In addition to indirect effects, we also predicted multiple cell cycle genes as direct p63 targets (Table 1). […] In addition to p63’s role in driving the expression of some cell cycle genes directly, the entire set of cell cycle genes may be subsequently up-regulated indirectly through p63’s pro-proliferative targets.”

2) Figure 9: Is the identified p63-signature differentially expressed in reference to the state of p63 (over expressed and/or amplified) in publicly available datasets of HNSC with expression and genomic data? Also, in Figure 9, is the identified signature prognostic specifically in HNSC and not in other SCC types?

We certainly believe that the reviewer is raising two critical questions here. In the following, however, we would like to argue that this matter’s complexity is best addressed in independent studies.

In the context of the originally submitted manuscript, we had not explored the correlation between p63 expression and levels of the 28-gene set. HNSC was chosen because of its known dependence on p63 and the availability of a large dataset via the TCGA. The purpose of this brief analysis was to test and showcase the potential of the genes that we identified as being frequently up-regulated by p63. Therefore, already in the original manuscript version we only concluded that “this 28-gene set may contain particularly promising candidates for therapeutic interventions and for the use as biomarkers”. Consequently, we noted that, while interesting, “this finding calls for a more detailed assessment of ubiquitous p63/SCC genes as biomarkers in the future.” (Results section).

To further clarify that we did not identify a signature for prognostic purposes, we added to the Discussion section:

“We provided a showcase example” and replaced ‘gene signature’ with ‘gene set’.

We indeed share the excitement about potential p63 target gene-based prognostic signatures for HNSC and other SCC types. Yet, the development of a meaningful prognostic gene signature requires a whole study, out of the scope of this work and not within our core expertise. To follow the questions of the reviewer nevertheless, we explored the relationship between the expression of the 28-gene set with p63 levels in the new Figures 8C, D, and Figure 8—figure supplement 1:

“Expression of the 28-gene set correlated positively with p63 expression when p63 expression was rather low (FPKM <20), but showed a saturation and no further correlation when p63 expression was high (FPKM >20; Figure 8C, D, and Figure 8—figure supplement 1). This indicates that p63 levels influence the 28-gene set in a switch-like manner where a saturation of p63-dependent activation is quickly reached in HNSC cells.“

3) To increase the long-term usability of this work for future studies, can the authors report the gene and the DNA sequence of the response element in the Table 3 as the reported position may change with the updating of the human genome sequence (GRCh38)?

Following the reviewers’ comment, we added the DNA sequence of the p63REs and the genes associated with the respective peaks to Supplementary file 3. Notably, to make our analysis contemporary all our data and subsequent analyses are already based on the current human genome version hg38/GRCh38.

4) The authors report that there are many more number of p63 binding sites in the genome as compared to the number of p53 binding sites. Is this because basal p63 levels are much higher than basal p53 levels? Can the authors comment on the relative molecules per cell of p63 and p53? Or is it possible that this difference is because p63 requires shorter sequences in the DNA and is assisted by a number of cofactors?

We thank the reviewer for this important remark. We have incorporated this matter in the description of our data. All p63 and p53 ChIP-seq data are derived from cells with high basal p63 and stimulated/induced p53 levels, respectively. Thus, even though basal p53 levels are low in most of these cells, they were activated/induced prior to the ChIP experiments. Our rationale is that both p63 and p53 levels are sufficiently high to occupy most relevant binding sites in the genome. Since the molecule numbers of p63 or p53 have not been assayed in any of the original studies, we are unfortunately unable to give any specifics. To clarify this, we added an explanatory sentence at the beginning of the respective Results paragraph:

“Notably, p63 and p53 data was collected from cells with strong basal p63 expression and stimulated p53 expression, respectively.”

Furthermore, our results suggest that p63 requires full-length recognition sequences, which, however, can deviate more strongly from the canonical motif as compared to p53. Our interpretation of the results is that p63’s ability to recognize a greater variety of motifs underlies its capacity to bind to many more sites in the genome as compared to p53. To clarify this towards the reader, we added each one additional sentence to the Results and Discussion sections:

“This broader binding repertoire most likely underlies p63’s capacity to engage with substantially more binding sites than does p53.”

“Notably, p63’s ability to bind to a greater variety of recognition motifs most likely underlies the markedly greater number of p63 compared to p53 binding sites in the genome.”